# Association of Image-Defined Risk Factors with Clinical, Biological Features and Outcome in Neuroblastoma

**DOI:** 10.3390/children9111707

**Published:** 2022-11-08

**Authors:** Balanescu Laura, Balanescu Radu, Cimpeanu Patricia, Moga Andreea

**Affiliations:** 1Department of Pediatric Surgery, “Grigore Alexandrescu” Clinical Emergency Hospital for Children, 011743 Bucharest, Romania; 2Department of Pediatric Surgery, “Carol Davila” University of Medicine and Pharmacy, 050474 Bucharest, Romania

**Keywords:** neuroblastoma, pediatric, surgery, prognosis

## Abstract

Background: Neuroblastoma (NB) is the most common pediatric extracranial solid tumor and the most common cancer encountered in children younger than 12 months of age. Localized tumors have a good prognosis, but some cases undergo treatment failure and recurrence. The aim of the study was to analyze the link between the neuroblastoma risk factors and the prognosis for patients diagnosed with NB. Method: All patients admitted to the department of Pediatric Surgery, “Grigore Alexandrescu” Clinical Emergency Hospital for Children, between 1 January 2010 and 1 July 2022 were included in this analysis when diagnosed with neuroblastoma. Results: Thirty-one patients with NB were admitted to the surgical department, 20 boys and 11 girls. We observed an association between large tumors and positive imaging-defined risk factor (IDRF) status; The Fisher test showed an association between the tumor’s diameter when bigger than 8 cm and a positive IDRF status, with *p* < 0.001. We supposed that positive IDRF status at diagnosis may be linked to other prognostic factors. We discovered that an NSE value over 300 was associated with IDRF status (*p* < 0.001, phi = 0.692) and death. Conclusions: This study confirms the impact of IDRF status at diagnosis as it can be clearly correlated with other risk factors, such as a high level of NSE, MYCN amplification status, large tumor size, incomplete tumor resection, and an unfavorable outcome.

## 1. Introduction

Neuroblastoma (NB) is the most common pediatric extracranial solid tumor encountered in children younger than 12 months of age [1,2]. Frequently, the tumor arises from the adrenal glands, but it can also be found anywhere along the paraspinal areas from the neck to the pelvis [3,4,5]. Forty percent to fifty percent of patients have metastatic disease or aggressive localized disease at the time of diagnosis [6]. The majority of children are diagnosed before the age of 5, with male patients being more frequently affected than females [7]. The clinical behavior of neuroblastomas is quite heterogeneous, with some cases presenting with spontaneous regression, while others have a rapid progression with an unfavorable outcome [8,9]. Most localized tumors present with a favorable outcome that may require just surgical resection, while some cases undergo treatment failure and recurrence. Due to the various clinical scenarios and outcomes, patients with neuroblastomas should benefit from accurate staging systems.

The International Neuroblastoma Staging System (INSS) was developed in 1986 and is based on the degree of resectability of the tumor, as well as certain pathological features of the tumor mass [10]. While this system was adopted in many countries, there are certain aspects that create a challenge when using it. For instance, the same tumors can be either stage 1 or 3, depending on the extent of the surgical excision, which is based on the surgeon’s experience, and as such, a direct comparison between different clinical trials is not feasible. Another limitation of the INSS is the correct assessment of lymph nodes, as the number of lymph nodes which are sampled during surgery is different depending on each individual surgeon. Moreover, the assessment of extra regional lymph nodes is difficult to evaluate across different studies [8,10,11,12].

In 2004, the International Neuroblastoma Risk Group (INRG) task force was formed in order to develop the INRG staging system (INRGSS) and the INRG risk classification system. The INRGSS can assess patients with neuroblastoma prior to surgery or any other treatment. Tumors are classified as either L1 or L2, depending on whether one or more imaging-defined risk factors (IDRFs) are present; these factors can predict the risk associated with tumor resection [13]. With the INRGSS system, the focus changed from surgical staging to imaging-based staging, as preoperative, diagnostic images are more easily reproducible than intraoperative findings, thus allowing for a more homogenous overview of the information gathered from different centers [14]. It is important to understand that the INRGSS is not meant to replace the INSS, as both systems should work in parallel. The IDRF and the INRGSS should be used in the beginning when establishing the initial diagnosis, as well as during treatment. The IDRFs are listed in Table 1 [10,11,12].

The prognosis of neuroblastoma is variable and has been proven to depend on tumor staging, patient age, DNA content, as well as tumor oncogenes. As such, the genetic landscape of neuroblastomas is important when determining the risk stratification for patients [8]. Twenty percent of patients with neuroblastoma present with MYCN amplification, and many studies have linked MYCN amplification and ploidy, together with tumor histopathology, with prognosis in these patients [15]. Tumor biopsy comes with risks such as bleeding or organ damage, while needle biopsy may not always be accurate, and as such, employing imaging characteristics is important in providing information for both primary and metastatic tumors [8,16]. Favorable and unfavorable subtypes are based on the level of Schwannian stroma present in the tumor and on the mitosis-karyorrhexis index (MKI) [17,18].

The association of positive IDRFs at the time of initial diagnosis with poor outcome is a subject of controversy, with many authors suggesting that the IDRF status could change after chemotherapy, and as such, a new evaluation should be performed right before surgery, as it may have a different prognosis [19,20,21,22].

The aim of our study was to analyze the link between neuroblastoma risk factors and prognosis in pediatric patients and to see if there is an association between IDRF and other prognostic factors such as age, MYCN status, NSE value and tumor size. 

## 2. Materials and Methods

We performed a retrospective review of all the charts of the patients who were admitted to our hospital with the diagnosis of neuroblastoma between 1 January 2010 and 1 July 2022. The use of the data from the medical database was approved by the hospital’s ethics committee. Electronical medical records and available charts were reviewed, including pathology and intraoperative reports. Forty patients with neuroblastoma were identified but nine patients were excluded from the study due to a lack of the essential medical information which was required for analysis. As such, thirty-one patients with neuroblastoma were included in our study. 

The IDRF status for each patient was evaluated, and we hypothesized that a positive IDRF status at diagnosis may be linked to other prognostic factors. We analyzed demographic, clinical, histological, and imagistic characteristics at the time of diagnosis, as well as during treatment, and patient outcome was evaluated. In order to establish the significance of several prognostic factors, we analyzed the associations and performed a statistical analysis using the Fisher test and the phi coefficient. The Fisher exact test was used to compare proportions and the phi coefficient was calculated in order to measure the associations. A ratio test *p*-value ≤ 0.05 was considered statistically significant. 

## 3. Results

From 2010 to 2022, thirty-one eligible patients with NB were admitted to the surgical department, twenty boys and eleven girls. The medical history was obtained and only six patients had a history of familial cancer. The median age at the time of diagnosis was 22 months, ranging from the newborn period to over 8 years of age. Of the thirty-one patients, thirteen were under 18 months of age (41.93%). 

Evaluation of the tumor mass was performed initially by ultrasonography (US), with subsequent computer tomography (CT) or magnetic resonance imaging (MRI). Extensive imaging was required for every neuroblastoma staging, and although ultrasonography was used for primary tumor detection due to having the advantage of being performed without sedation, cross-sectional imaging was essential in all cases. As such, for all our patients we either performed a CT scan or an MRI for accurate tumor evaluation. (Figure 1).

In order to establish an initial diagnosis, tumor markers such as neuron-specific enolase (NSE) and urinary levels of catecholamine metabolites, including vanillylmandelic acid (VMA) and homovanilic acid (HVA), were measured. NSE was obtained from all the patients and was abnormal in all cases, with a median value of 241.7 ng/mL. An increased level of lactate dehydrogenase (LDH) was also noted in all cases, with the mean value being 865.06 IU/L. Sixteen patients were found to present with anemia, while eighteen had an inflammatory syndrome. When looking at the neuroblastoma tumor markers, we found a strong association between an NSE value over 300, a positive IDRF status, and an unfavorable outcome (*p* < 0.001, phi = 0.692). No correlation was found between the LDH level and IDRG positive status or an unfavorable outcome. 

We also looked at the pretreatment blood counts to possibly identify whether the neutrophil–lymphocyte ratio (NRL), lymphocyte count, neutrophile count, platelet count, or platelet–lymphocyte (PLR) ratio had a prognostic value for pediatric patients with neuroblastoma. The median NRL was 0.45, while the mean PLR was 47.96. When looking into a possible association between the pretreatment blood test values with IDRF positive status and unfavorable outcome, we found no statistically significant correlation. 

The patients were classified according to both the INSS and the INRF staging systems. Based on the INSS classification system, we found seven cases of stage 1 and stage 2 and twenty-four cases of stage 3 and stage 4/4 s, while according to the INRF staging system, we had six cases of L1, six cases of L2, nine cases of M and ten cases of MS. As previously stated in the literature, IDRF status at the time of diagnosis is an important prognostic factor. Using the Fisher test, we found a strong association between positive IDRF status and unfavorable outcome (*p* = 0.002), with phi = 0.571. (Table 2)

The IDRF status of the primary tumor when the initial diagnosis was established was evaluated as either positive or negative. There were sixteen cases of positive IDRF and fifteen cases of negative IDRF. We looked at tumor size for all the patients and found a median tumor size of 9 cm in diameter with a minimum of 2.3 cm and a maximum of 20.3 cm. According to the Fisher test, when correlating tumor size and IDRF status, we found that a tumor diameter over 8 cm was associated with a positive IDRF status (*p* < 0.001). We also analyzed the link between the size of the tumor and the long-term outcome and found that all the deceased patients presented with a large, bulky mass with an average of 12 cm, with a minimum of 9 cm, and a maximum of 20.3 cm (*p* = 0.027). 

In ten cases biopsy was performed prior to initiation of the chemotherapy protocol, while for the rest of the patients, the histopathology of the tumor was analyzed after tumor resection. The histopathological exam revealed five undifferentiated NBs, nineteen poorly differentiated NBs, and one differentiating NB; six cases had no histopathological result, a positive diagnosis being established just by using biological and radiological criteria. Amplification of MYCN was expressed in eleven cases, while in fourteen cases no amplification was found. The Fisher test also proved an association between MYCN amplification and positive IDRF status, with *p* = 0.004 and phi = 0.623.

Nineteen patients presented with metastatic disease, with a predominance of ganglionic involvement and liver metastasis (Table 3).

Treatment was initiated in twenty-eight cases after diagnosis and included chemotherapy, surgery, radiation, stem cell transplant, and retinoid therapy. The treatment plan for each patient was established by a multidisciplinary team consisting of a pediatric oncologist, pediatric surgeon, and radiologist. The necessity for neoadjuvant treatment was established based on the possibility of the removal of the entire tumor. We analyzed the general choices of treatment and found nine cases of upfront resection, fourteen cases of surgical treatment after neoadjuvant therapy, and eleven cases that did not benefit from surgical resection. When looking at the possibility of tumor removal, we found a link between IDRF status and surgical treatment. Positive IDRF was associated with incomplete tumor resection (*p* = 0.002, phi = 0.612). 

Neoadjuvant chemotherapy was necessary in seventeen cases, and two patients died from complications and did not benefit from surgical treatment. Six patients did not receive chemotherapy, while nine patients followed a postoperative chemotherapy regime.

We analyzed the type of surgical approach that was chosen for both biopsy or tumor excision and found that in five cases a minimally invasive approach was favored. No correlation was established between complete resection and the type of approach. Complete resection was obtained in twelve cases and an incomplete excision was performed in eight cases. There was no association found between positive IDRF and local relapse (*p* = 0.083) or adjacent organ resection (*p* = 1). Despite extensive preoperative planning, in seven cases, in order to obtain complete tumor excision, adjacent organ resection was also required. Two patients presented with intraoperative complications, such as extensive bleeding, and reintervention was required in eight cases. We hypothesized that an incomplete tumor resection may be a bad prognostic factor, but when we analyzed the impact of surgical treatment, we found no significant association between the type of excision or local relapse and death. Incomplete tumor resection may influence overall survival from the time of diagnosis, but the data are not sufficient to sustain this theory at this time. A larger group of patients is required to further investigate this possible association. 

When looking at histopathological tumor characteristics and the operative approach, we found no association between MYCN status and complete or incomplete excision (*p* = 0.356). Moreover, MYCN status could not be correlated with local relapse (*p* = 0.350). 

Although chemotherapy and surgery are the most effective treatment options, seven patients had autologous stem cell transplantation and eight patients needed retinoid therapy. The most popular chemotherapy drugs were vincristine (n = 22), carboplatin (n = 23), irinotecan (n = 21), and cyclophosphamide (n = 15). The number of chemotherapy sessions varied, with a mean of six sessions per patient (minimum = 1, maximum = 16). Seventeen patients had a satisfactory chemotherapy tolerance. (Table 4)

Complications were noted in 21 cases: renal disfunction (n = 5), liver disfunction (n = 6), respiratory distress (n = 5), infection (n = 3), thrombocytopenia (n = 16), ascites (n = 4), hemorrhage (n = 6), disseminated intravascular coagulation (n = 1), and death (n = 8). Local relapse was observed in seven cases. Although most patients had high-stage neuroblastoma, overall survival at 2 years after the initial diagnosis was 77.42%. All seven patients who presented with an unfavorable outcome were IDRF positive. Six patients were initially diagnosed with a stage M or MS according to the INRG staging system, and one was an L2. These patients’ ages ranged from 13 months to 99 months with a median of 22 months. There were no deaths among the newborns. An unexpected surprise was the strong association between MYCN status and unfavorable outcome (*p* = 0.003, phi = 0.634). All the deceased patients presented with an MYCN amplification. Thus, this proves that MYCN amplification is a strong prognostic factor for the outcome in patients with neuroblastoma. 

## 4. Discussion

The available literature shows that 90% of the cases of patients with neuroblastoma are diagnosed before the age of 5, with 30% of cases being identified within the first years of life. These data are similar to those found in our study, with 83.87% of patients (n = 26) being under 5 years of age at the time of diagnosis and 29.03% under the age of 1 (n = 9). The median age for our patients is similar to that in other published studies (22 months). We also found a higher incidence in male patients (male: female ratio 1.8:1), which is similar to the other available data. Family history was difficult to obtain, and although six patients had a history of familial cancer, the data were poor, and we could not establish a link with other specific conditions [23,24,25,26,27,28].

Our results confirm that the most common primary site of NB is the adrenal gland, followed by extra-adrenal abdominal lesions (22.58%) and mediastinum lesions (3.1%). There were no cases of primary neck lesions. When comparing our results to those previously published, we observed a slightly greater preponderance of the adrenal glands origin (74.19% versus 35–65%) [1,3,23,24,25].

The size of the tumor mass was defined as the maximum diameter of the tumor and should be considered as a predictive factor, as has been reported for other types of cancer, such as lung cancer or gastric cancer [29]. Our data revealed an association between the tumors over 8 cm in diameter and positive IDRF status, amplified MYCN, and poor outcome. The cut-off value for our study was larger when compared to the results published in other studies (8 cm versus 4 cm). We believe that to be case as most of our patients presented with large tumors at the time of diagnosis. To better evaluate the effectiveness of tumor size as a potential risk factor, a larger group of patients is required [30].

Most commonly, neuroblastomas can present with bone or bone marrow metastases at the time of initial diagnosis, but in our case, the liver was the most popular site for distant lesions, followed by bone and skin/subcutaneous tissue [15,26]. 

Although cutaneous lesions are rare among metastases, it may be the primary reason why parents bring the child to the hospital, as has been seen in other studies [25,31]. Tumors can metastasize either via the lymphatic or the hematogenous systems. A tumor that has spread to the ganglia outside the cavity of origin still represents a metastatic lesion [20]. A study from the Vanderbilt University Medical Center in Nashville proved that up to 50% of neuroblastomas spread to lymph nodes, a value which is similar to our findings (n = 15, 48.38%) [32].

When diagnosed with cancer, the questions are the same: how bad is the cancer and how bad is it going to become? In order to answer these questions, we need predictive factors and evolutive links. Individual variability still remains unpredictable, but for the majority of cases, we need answers, which usually come in the form of statistical results. 

The diagnostic protocol for a neuroblastoma includes tumor markers, ultrasonography followed by CT scan or MRI imaging, and histopathological analysis when that is possible. Differential diagnosis for these patients usually includes pheochromocytoma, Wilms’ tumor, or rhabdomyosarcoma, and the histopathological result is extremely important for treatment. We analyzed the NSE levels for all our patients and found them to be abnormal (normal value < 17 ng/mL) in all cases. An NSE value over 300 ng/mL was associated with positive IDRF status and a poor outcome. The cut-off value for association with IDRF status was 200 ng/mL. Many publications revealed that an NSE concentration in serum of over 100 ng/mL can affect survival [33,34]. Other studies discouraged the use of serum NSE for screening or diagnosis of NB due to the existence of possible confounding factors that may cause a false-positive increase, but we believe that consistent high values should have a predictive role [33]. 

Molecular testing of the tumor has become important for overall prognosis, and MYCN gene amplification is an important marker for poor prognosis and aggressive disease in neuroblastoma. Neuroblastoma cells may suffer genetic damage; the MYCN gene is a proto-oncogene that is amplified in more than 10 copies in 20–30% of cases, and it was associated with aggressive disease and poor outcome [1,3,7,16]. We found eleven cases (35.48%) with amplified MYCN. We analyzed multiple associations and found a link between MYCN status and unfavorable outcome. As seen in other studies, the MYCN amplification is a prognostic factor and is usually associated with poorly differentiated or undifferentiated neuroblastoma [4,18,25,34]. Our data do not support this theory, but we did observe a link between the MYCN amplification and positive IDRF status, which may suggest that an unfavorable outcome can be sustained by many factors even if the histology is not on the same side. This theory was also tested in a study by Temple W.C. in 2020 [8]. MYCN amplification was used for risk stratification and treatment guidelines [35,36,37]. Nevertheless a study from South Korea proved that the reduction in tumor volume and the reduction in serum NSE levels were greater after chemotherapy in an MYCN-amplified tumor, meaning that the tumor’s response was better [36]. 

Imaging modalities such as CT and MRI are important for the diagnosis and presurgical assessment of tumor resectability [37,38]. Surgical treatment is recommended in selected cases, based on location and tumor infiltration [27]. Every case is analyzed by a multidisciplinary team (oncologist, surgeon, and radiologist) in order to establish whether the patient is a candidate for up-front surgery or for neoadjuvant chemotherapy. The aim of the surgical intervention in these cases is to obtain a complete resection, but the best timing for surgery is definitively a strategic decision [6]. 

In our study, the quality of resection was obtained from surgical reports, not from postoperative imaging. We found that positive IDRF status at diagnosis means incomplete resection and is strongly correlated with an unfavorable outcome. In order to obtain complete resection, adjacent organ resection was needed in seven cases (22.58%). Many authors consider that surgery has better results after chemotherapy, when the tumor size is reduced, and as such, the chance of resectability increases, which leads to a more favorable outcome [12,39,40]. Big tumors have a positive IDRF status and are associated with a poor prognosis. 

In some cases, when the tumor is completely removed, the encasement of major retroperitoneal or mediastinal structures might be linked to microscopically positive margins [40]. Nevertheless, many studies encourage surgical treatment in spite of the positive margins, and the instantaneous removal of a large quantity of tumor cells may improve the outcome [19]. A study from Japan proved that complete resection is not always required; selected cases of microscopically incomplete resection had a better survival rate than expected [41]. A cohort study that included high-risk neuroblastomas proved that surgical resection (over 95% of tumor) improved survival, but the timing of surgery remains a strategic tool [41,42]. For this category of patients, local control is better if surgery is performed after induction chemotherapy [6]. 

Some authors recommend neoadjuvant chemotherapy in the presence of IDRFs [43]. The presence of IDRFs can predict the extent of the surgery, and it should be integrated as part of the treatment plan, but it is probably best used in association with other prognosis factors, such as MYCN amplification, NSE level, tumor size, and histological characteristics [20,21,42,44,45,46]. Our results are similar to a systematic review of 3725 studies that concluded that IDRF-positive neuroblastomas have a higher risk of incomplete surgical resection and a higher risk of 5-year mortality or relapse [47]. 

Some authors have suggested that IDRF status should change prior to surgery, after chemotherapy, but this is still a point of controversy in the management of neuroblastoma patients [45,48,49]. Our data showed a statistically significant correlation between positive IDRF status and an incomplete tumor resection, but no correlation was found between incomplete surgical resection and overall survival.

In our group, all the biopsies were performed either by laparotomy or laparoscopy. Fine-needle aspiration cytology was not used in our clinic in order to guarantee enough biopsy material for histological and immunohistochemistry tests [50]. In our center, minimally invasive surgery (MIS) is usually an option for biopsy but based on the SIOPEN guidelines considering MIS for tumor resection, where the presence of IDRF is an absolute contraindication for MIS, many surgeons do not choose this approach, unless it is proven to be safe and feasible [51,52,53,54]. 

Neuroblastoma is a type of cancer with an extremely variable evolution, from the newborn period with spontaneous regression to aggressive tumors that keep growing despite multimodal treatment [55,56]. Modern treatment for neuroblastoma is based on diagnosis and prognosis, with risk classification being used to conduct the treatment strategies. The INRG risk stratification system has not yet been adopted in all centers, but it has proven to be a valuable tool that will help to optimize neuroblastoma therapy models [57].

Despite that, many high-risk neuroblastoma patients respond poorly to treatment, and long-term survival is less than 50% [56,57]. Our data revealed a 2-year overall survival of 77.42%. 

## 5. Conclusions

This study confirms the impact of IDRF status at diagnosis as it can be clearly correlated with other risk factors, such as a high level of NSE, MYCN amplification status, large tumor size, incomplete tumor resection, and an unfavorable outcome. 

## Figures and Tables

**Figure 1 children-09-01707-f001:**
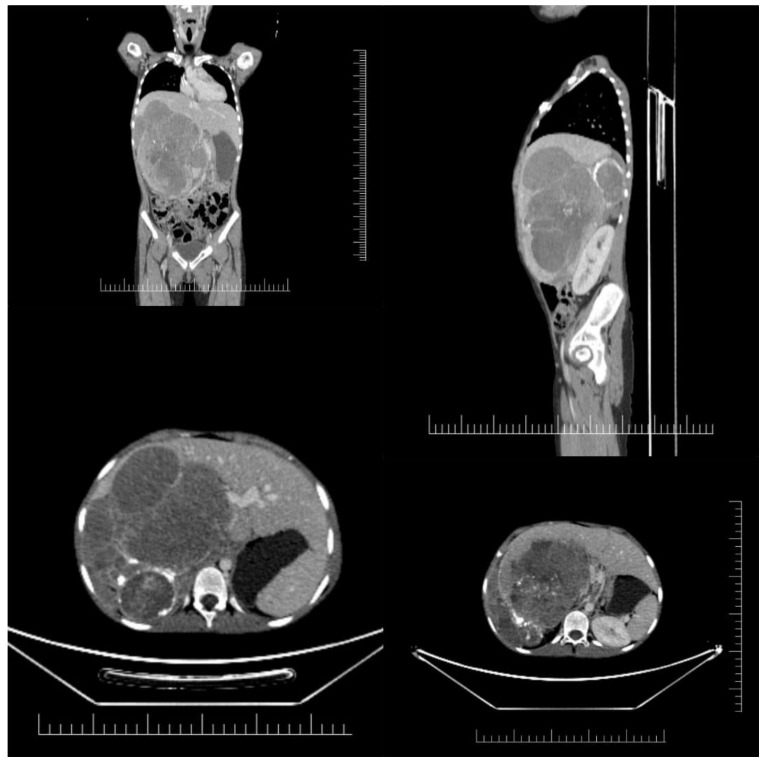
CT scan showing a bulky abdominal neuroblastoma with liver invasion and encasement of abdominal aorta and inferior vena cava.

**Table 1 children-09-01707-t001:** Definitions of image-defined risk factors (IDRF groups) [1,11].

Anatomic Region	Description
**Different body compartments**	Ipsilateral tumor extension within two body compartments (neck and chest or pelvis and abdomen
**Cervical region**	Tumor encasing internal jugular vein, the carotid artery, or the vertebral artery Extension of tumor to skull base Compression of the trachea
**Thoracocervical Junction**	Tumor encasing the brachial plexus roots Tumor encasing carotid artery, subclavian vessels, vertebral artery Compression of the trachea
**Thorax**	Tumor encasing the aorta or important branches Tumor compressing the trachea/bronchi Lower mediastinal tumor with costovertebral junction infiltration (T9 and T12)
**Thoracoabdominal junction**	Tumor encasing the vena cava or the aorta
**Pelvis or Abdomen**	Tumor infiltrating the hepatoduodenal ligament or the porta hepatis Tumor encasing branches of the superior mesenteric artery Tumor encasing the origin of celiac axis or of the superior mesenteric artery Invasion of one or both renal pedicles Tumor encasing the vena cava or the aorta Tumor encasing the iliac vessels Pelvic tumor crossing the sciatic notch
**Infiltration of adjacent structures**	Diaphragm, pericardium, liver, kidney, mesentery, and the duodenopancreatic block
**Intraspinal tumor extension**	Invasion of more than 1/3 of the spinal canal, non-visible perimedullary leptomeningeal spaces or abnormal signal intensity of the spinal cord

**Table 2 children-09-01707-t002:** Demographic, clinical, and biological results.

Characteristic	No.
**Sex**	N = 31
**Male**	20
**Female**	11
**Primary tumor site**	
**Neck**	0
**Chest**	1
**Abdomen**	7
**Adrenal**	23
**Abnormal NSE at diagnosis**	31
**MYCN status**	
**Non-amplified**	14
**Amplified**	11
**Not specified**	6
**INSS stage**	
**½**	7
**¾**	24
**INRGSS stage**	
**L1**	6
**L2**	6
**M**	9
**MS**	10
**IDRF**	
**Positive**	16
**Negative**	13

**Table 3 children-09-01707-t003:** Metastasis.

Characteristic	No. (n = 31)
**Pulmonary metastasis**	1
**Hepatic metastasis**	12
**Bone metastasis**	7
**Bone marrow**	5
**Skin/soft tissue metastasis**	4
**Ganglia metastasis**	15
**Renal metastasis**	2

**Table 4 children-09-01707-t004:** Treatment.

Surgical Treatment	N = 31
**Upfront resection**	**9**
**After neoadjuvant treatment**	**14**
**No excision**	**11**
**Biopsy**	**10**
**Complete excision**	12
**Incomplete excision**	8
**Upfront chemotherapy**	17
**Type of intervention**	
**MIS**	5
**Open**	17
**Adjacent organ resection**	7
**Reintervention**	8
**Autologous stem cell transplantation**	7
**Retinoid therapy**	8

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
