# Peer review of "Association of Image-Defined Risk Factors with Clinical, Biological Features and Outcome in Neuroblastoma"

_children, 2022, doi:10.3390/children9111707_

Round 1
Reviewer 1 Report
Dear author
Congrats on your work. The article contains a lot of useful information on the issue and the topic is very interesting, However, some revisions are necessary
Author Response
thank you for your feedback. We have edited the article and are looking forward to hearing your thoughts.
best wishes!
Reviewer 2 Report
The authors are on the right track in looking at MRI and CT, NSE and MYC-N characteristics that predict outcome and treatment choice of neuroblastoma. However the manuscript is massively disorganized, filled with typographic and formatting errors, bad grammar, repetitions, omissions, and missing important data. This manuscript should not be accepted by any journal in its present form. Professional medical editing and organization of presentation will be required.
An ethical concern with this paper: in most jurisdictions informed consent of individuals is not required for retrospective chart review when numbers are large enough such that individuals can’t be identified. If indeed individual informed consent was obtained for all patients then the usual statements must be listed- what the informed consent covered and how many declined. It is unlikly all -as the authors state- agreed.
I suggest the authors recheck this matter and state that their study was exempted from individual informed consent by their local IRB that approved their chart review if indeed this was the case.
More recent background reference papers should be used instead of several outdated ones the authors used. Two examples listed below.
In their revision I would require listing selected CBC values, including the NLlR, presenting signs and symptoms, and correlating these parameters with the individual entries on their eight point IDRF list. I would also require 2 year OS % listed for all eight IDRF positives, correlations with OS and all lab parameters and OS as function of presenting sign/symptom. Although I strongly recommend rejecting this work, I would also encourage the authors to correct the numerous errors, reorganize their presentation, add the lab value and other correlations, and then resubmit. This work does have potential to further our understanding and treatment of neuroblastoma, but not so in its present form.
1: Sokol E, Desai AV. The Evolution of Risk Classification for Neuroblastoma. Children (Basel). 2019 Feb 11;6(2):27. doi: 10.3390/children6020027. PMID:30754710; PMCID: PMC6406722.
2: Van Arendonk KJ, Chung DH. Neuroblastoma: Tumor Biology and Its Implications for Staging and Treatment. Children (Basel). 2019 Jan 17;6(1):12. doi:10.3390/children6010012. PMID: 30658459; PMCID: PMC6352222.
3: Gomez RL, Ibragimova S, Ramachandran R, Philpott A, Ali FR. Tumoral heterogeneity in neuroblastoma. Biochim Biophys Acta Rev Cancer. 2022 Sep 23:188805. doi: 10.1016/j.bbcan.2022.188805. Epub ahead of print. PMID: 36162542. 4: Lundberg KI, Treis D, Johnsen JI. Neuroblastoma Heterogeneity, Plasticity, and Emerging Therapies. Curr Oncol Rep. 2022 Aug;24(8):1053-1062. doi: 10.1007/s11912-022-01270-8. Epub 2022 Apr 1. PMID: 35362827; PMCID: PMC9249718.
Author Response
The authors are on the right track in looking at MRI and CT, NSE and MYC-N characteristics that predict outcome and treatment choice of neuroblastoma. However the manuscript is massively disorganized, filled with typographic and formatting errors, bad grammar, repetitions, omissions, and missing important data. This manuscript should not be accepted by any journal in its present form. Professional medical editing and organization of presentation will be required. – the structure of the manuscriped was changed as per your observations and typing errors as well as gramatical ones were changed
An ethical concern with this paper: in most jurisdictions informed consent of individuals is not required for retrospective chart review when numbers are large enough such that individuals can’t be identified. If indeed individual informed consent was obtained for all patients then the usual statements must be listed- what the informed consent covered and how many declined. It is unlikly all -as the authors state- agreed.
I suggest the authors recheck this matter and state that their study was exempted from individual informed consent by their local IRB that approved their chart review if indeed this was the case. – This is a fair observation and I think it was actually a misunderstanding in the way we initially expressed the matter. The hospital’s ethichs committee gave their consent for us to use the data we found in charts and in the database. No individual consent was obtained from patients regarding this study.
More recent background reference papers should be used instead of several outdated ones the authors used. Two examples listed below – we did use some of the suggested articles as reference
In their revision I would require listing selected CBC values, including the NLlR, presenting signs and symptoms, and correlating these parameters with the individual entries on their eight point IDRF list. I would also require 2 year OS % listed for all eight IDRF positives, correlations with OS and all lab parameters and OS as function of presenting sign/symptom. Although I strongly recommend rejecting this work, I would also encourage the authors to correct the numerous errors, reorganize their presentation, add the lab value and other correlations, and then resubmit. This work does have potential to further our understanding and treatment of neuroblastoma, but not so in its present form.
- we went back to our initial data and as suggested looked at the NLIR of our patient, but no association was found. We also looked at the 2 year OS which we mentioned in our results.

Reviewer 3 Report
Association of image-defined risk factors with clinical, biological features and outcome in neuroblastoma
It would be good if the authors could do a comprehensive language and grammar check since I have found multiple issues here and there.
Please refrain from using abbreviations that are not defined in the abstract. Assume the reader does not know what it stands for.
“We discovered that NSE value over 300 was associated with IDRF status (p<0.001, Phi= 0.692) and death. “
The association between IDRF (linked to INRG staging) and clinicopathological attributes are apparently what the paper is testing. Please expand on what is already known in the field on this topic.
Please readdress your method of citation, since I don’t think this complies with the MDPI style.
We analysed the link between neuroblastoma risk factors and prognosis; we included age, MYC-N status, NSE value, Tumor size, image-defines risk factors (IDRFs), and histopathological results. MYCN needs to be corrected, NSE needs to be defined; Neuron-specific enolase (NSE).
The introduction suddenly and abruptly stops and any narrative is replaced by a huge table. Please rework your introduction and build momentum about the study at hand, what the known aspects are and what aspects are unknown, and then state your objective. Your introduction is not acceptable in its current form.
Please provide details of the ethical approval of the study in detail and any criteria for patient selection.
The ratio rest p-value for inclusion was p ≤ 0.05. Rest?
It is not clear on what grounds the patients were categorised or if indeed they were categorised. What selection process or criteria was used? Perhaps stage?
Did the authors do any classification at all or did they test for the tumour size and some clinical attributes and based on that look at association with IDRF? It is not clear what the study design is, I think the tumour size was an important factor that the authors have considered and then the authors move on to establishing the association between IDRF and other clinical attributes. Does a larger tumour correlate with risk or is risk linked to stage that per se is a factor of the anatomical location and dissemination of the tumour? How can tumour size directly impact risk?
The link between tumour size, IDRF, and risk (if any) needs to be explained.
We also analysed the link between tumor’s size and deceased patients and found that all patients who died had big, bulky tumors with an average of 12 cm (minimum=9 cm, maximum 20.3 cm). How about other characteristics such as MYCN characteristics, stage, ploidy, chromosomal alterations and other factors that contribute to risk? I can see that was mentioned in the next section. However, the authors could do a better job of linking these characteristics together or they could give a table with detailed information about the 31 tumours, so the authors can have a better way of visualizing this information. I appreciate what you have done in tables 2 and 3 but I rather refer to the full characteristics of the 31 tumours that could be presented in a table or supplementary file.
Also, the presurgical staging system (the INRG staging system) was developed using image-defined risk factors (IDRF) and INSS. This looks like something that should have been in the introduction to build momentum to the question at hand, it is not well-placed here.
Then the authors move on to discuss the treatment options obtained by the 31 patients. Was there a link between treatment and IDRF? A whole section has been provided about treatment but I don’t see the point of it if the various treatment options in table 4 have not been comprehensively linked to IDRF (apart from surgical treatment).
In order to establish the significance of several prognostic factors, we analysed many associations and made a statistical analysis using Fisher’s test and Phi coefficient. What is not clear is why is the link between IDRF with risk/ prognostic factors significant, please make this clear.
IDRF status was the subject we concentrated on. We believe that it can be an important prognostic factor and it may be linked to other prognostic factors. Again, the kind of thing that should have been mentioned in the intro.
The data presented in the last section of the results (lines 140- 153), could go in a table for easier reading.
Our data revealed an association between tumors over 8 cm diameter and positive IDRF status, amplified MYCN, and poor outcome (death). The cut-off is bigger compared to literature results (8 cm versus 4 cm), but it remains to extend this analysis to a larger group of patients. INRG staging is based on IDRF, so this study claims that the size of the tumour is linked to INRG staging and also to other risk factors. Is that a fair conclusion of what this study is about? If so, why is it important that the size of tumour may play a role in staging and risk?
Our study supports previous publications, but the association of positive IDRF at diagnosis with poor outcome was a subject of controversy, many others suggested that IDRF status could change after chemotherapy, and reconsidering it right before surgery may have a different prognosis. In my opinion, this sentence is better suited to the introduction, since it is not clear to a reader, until this late point in the manuscript, if this angle was a matter of dispute warranting more investigation, so that the current study was justified to address this gap, hence the structure of the manuscript needs improvement. To be honest, understanding what your study was about, took some detective work and this should be improved.
The authors could also add a section on the effect of the implication of this finding and how it is important to the NB scientific community since this has not been clearly explained.
Please provide some future directions.
Author Response
Association of image-defined risk factors with clinical, biological features and outcome in neuroblastoma
It would be good if the authors could do a comprehensive language and grammar check since I have found multiple issues here and there. – thank you for your feedback. We hope that the newesr version if free of language and grammar errors.
Please refrain from using abbreviations that are not defined in the abstract. Assume the reader does not know what it stands for. – we tried to correct this oversight. Thank you for pointing it out to us!
“We discovered that NSE value over 300 was associated with IDRF status (p<0.001, Phi= 0.692) and death. “
The association between IDRF (linked to INRG staging) and clinicopathological attributes are apparently what the paper is testing. Please expand on what is already known in the field on this topic. – we tried to give a more comprehensive view of this matter in the introduction, as to make a better setting for the purpose of our article.
Please readdress your method of citation, since I don’t think this complies with the MDPI style. – the citing method was changed as per MDPI regulations.
We analysed the link between neuroblastoma risk factors and prognosis; we included age, MYC-N status, NSE value, Tumor size, image-defines risk factors (IDRFs), and histopathological results. MYCN needs to be corrected, NSE needs to be defined; Neuron-specific enolase (NSE).
The introduction suddenly and abruptly stops and any narrative is replaced by a huge table. Please rework your introduction and build momentum about the study at hand, what the known aspects are and what aspects are unknown, and then state your objective. Your introduction is not acceptable in its current form. – the introduction was re-written as to make it more suitable for the purpose of our study. m
Please provide details of the ethical approval of the study in detail and any criteria for patient selection. – ethical approval was obtained from the ethical board in our hospital. Patient criteria was also better described, as per your suggestion.
The ratio rest p-value for inclusion was p ≤ 0.05. Rest? – test, it was an unfortunate oversight!!!
It is not clear on what grounds the patients were categorised or if indeed they were categorised. What selection process or criteria was used? Perhaps stage? Did the authors do any classification at all or did they test for the tumour size and some clinical attributes and based on that look at association with IDRF? It is not clear what the study design is, I think the tumour size was an important factor that the authors have considered and then the authors move on to establishing the association between IDRF and other clinical attributes. Does a larger tumour correlate with risk or is risk linked to stage that per se is a factor of the anatomical location and dissemination of the tumour? How can tumour size directly impact risk?
The link between tumour size, IDRF, and risk (if any) needs to be explained.
We also analysed the link between tumor’s size and deceased patients and found that all patients who died had big, bulky tumors with an average of 12 cm (minimum=9 cm, maximum 20.3 cm). How about other characteristics such as MYCN characteristics, stage, ploidy, chromosomal alterations and other factors that contribute to risk? I can see that was mentioned in the next section. However, the authors could do a better job of linking these characteristics together or they could give a table with detailed information about the 31 tumours, so the authors can have a better way of visualizing this information. I appreciate what you have done in tables 2 and 3 but I rather refer to the full characteristics of the 31 tumours that could be presented in a table or supplementary file. – we tried to better explain our resoning and modified the format of our results so that things have a better flow to them.
Also, the presurgical staging system (the INRG staging system) was developed using image-defined risk factors (IDRF) and INSS. This looks like something that should have been in the introduction to build momentum to the question at hand, it is not well-placed here – we did as suggested and moved this statement
Then the authors move on to discuss the treatment options obtained by the 31 patients. Was there a link between treatment and IDRF? A whole section has been provided about treatment but I don’t see the point of it if the various treatment options in table 4 have not been comprehensively linked to IDRF (apart from surgical treatment). In order to establish the significance of several prognostic factors, we analysed many associations and made a statistical analysis using Fisher’s test and Phi coefficient. What is not clear is why is the link between IDRF with risk/ prognostic factors significant, please make this clear. – we tried to make things more comprehensible by re-writing the results section.
IDRF status was the subject we concentrated on. We believe that it can be an important prognostic factor and it may be linked to other prognostic factors. Again, the kind of thing that should have been mentioned in the intro. - we did as suggested and moved this statement and moved this statement
The data presented in the last section of the results (lines 140- 153), could go in a table for easier reading. Our data revealed an association between tumors over 8 cm diameter and positive IDRF status, amplified MYCN, and poor outcome (death). The cut-off is bigger compared to literature results (8 cm versus 4 cm), but it remains to extend this analysis to a larger group of patients. INRG staging is based on IDRF, so this study claims that the size of the tumour is linked to INRG staging and also to other risk factors. Is that a fair conclusion of what this study is about? If so, why is it important that the size of tumour may play a role in staging and risk? – it is hard to explain why the cut-off was higher in our study compared to other, we could only surmise that it is linked to the fact that most of our patients presented with bulky tumors ar the time of initial diagnosis.
Our study supports previous publications, but the association of positive IDRF at diagnosis with poor outcome was a subject of controversy, many others suggested that IDRF status could change after chemotherapy, and reconsidering it right before surgery may have a different prognosis. In my opinion, this sentence is better suited to the introduction, since it is not clear to a reader, until this late point in the manuscript, if this angle was a matter of dispute warranting more investigation, so that the current study was justified to address this gap, hence the structure of the manuscript needs improvement. – this is a fair point, and one we tried to adress when re-writing thr article.
To be honest, understanding what your study was about, took some detective work and this should be improved.
The authors could also add a section on the effect of the implication of this finding and how it is important to the NB scientific community since this has not been clearly explained.
Please provide some future directions.

Round 2
Reviewer 1 Report
It's ok with me.
Author Response
Thank you for your feedback!
Reviewer 2 Report
The goal of the authors, to present a predictive scoring system for pediatric neuroblastoma, is an important project. The IDRF status has good potential to benefit treatment planning and our research efforts. It should be widely known, used, and refined. Publishing the author's data will advance that goal.
Author Response
Thank you again for your feedback and response. We hope to have edited the manuscript in accordance to your suggestions.
Reviewer 3 Report
The authors have addressed my comments.
Author Response
thank you for your feedback!